# Effect of Al_2_O_3_ on Nanostructure and Ion Transport Properties of PVA/PEG/SSA Polymer Electrolyte Membrane

**DOI:** 10.3390/polym14194029

**Published:** 2022-09-26

**Authors:** Hamdy F. M. Mohamed, Esam E. Abdel-Hady, Mostafa M. Y. Abdel-Moneim, Mohamed A. M. Bakr, Mohamed A. M. Soliman, Mahmoud G. H. Shehata, Mahmoud A. T. Ismail

**Affiliations:** Physics Department, Faculty of Science, Minia University, Minia 61519, Egypt

**Keywords:** polymer electrolyte membrane, positron annihilation lifetime, hole volume, fuel cell, cross-linked PVA, Al_2_O_3_, proton conductivity, water uptake

## Abstract

Polymer electrolyte membrane (PEM) fuel cells have the potential to reduce our energy consumption, pollutant emissions, and dependence on fossil fuels. To achieve a wide range of commercial PEMs, many efforts have been made to create novel polymer-based materials that can transport protons under anhydrous conditions. In this study, cross-linked poly(vinyl) alcohol (PVA)/poly(ethylene) glycol (PEG) membranes with varying alumina (Al_2_O_3_) content were synthesized using the solvent solution method. Wide-angle X-ray diffraction (XRD), water uptake, ion exchange capacity (IEC), and proton conductivity were then used to characterize the membranes. XRD results showed that the concentration of Al_2_O_3_ affected the degree of crystallinity of the membranes, with 0.7 wt.% Al_2_O_3_ providing the lowest crystallinity. Water uptake was discovered to be dependent not only on the Al_2_O_3_ group concentration (SSA content) but also on SSA, which influenced the hole volume size in the membranes. The ionic conductivity measurements provided that the samples were increased by SSA to a high value (0.13 S/m) at 0.7 wt.% Al_2_O_3_. Furthermore, the ionic conductivity of polymers devoid of SSA tended to increase as the Al_2_O_3_ concentration increased. The positron annihilation lifetimes revealed that as the Al_2_O_3_ concentration increased, the hole volume content of the polymer without SSA also increased. However, it was densified with SSA for the membrane. According to the findings of the study, PVA/PEG/SSA/0.7 wt.% Al_2_O_3_ might be employed as a PEM with high proton conductivity for fuel cell applications.

## 1. Introduction

A fuel cell is a type of device that uses electrochemical reactions to transform the chemical energy of a fuel directly into electricity. A fuel cell looks like a battery in many ways, yet it can supply electrical energy over a significantly longer period. This is on the grounds that an energy component is consistently provided with fuel and air (or oxygen) from an outer source, while a battery contains just a restricted measure of fuel material and oxidants that are drained with use. Consequently, fuel cells have been used in space tests, satellites, and monitored rockets for quite a long time. The proton exchange membrane fuel cell (PEMFC) is one of the most advanced fuel cell designs. Hydrogen gas under pressure is constrained through a catalyst, commonly made of platinum, on the anode (negative) side of the fuel cell. At this catalyst, electrons are taken from the hydrogen atoms and conveyed by an outside electric circuit to the cathode (positive) side. The positively charged hydrogen ions (protons) then, at that point, go through the polymer electrolyte membrane to the catalyst on the cathode side, where they respond with oxygen and the electrons from the electric circuit to form water vapor (H_2_O) and heat. PEMFC uses a special electrolyte membrane that conducts protons between the anode and cathode. It has a higher power density, lower weight, and smaller volume than other fuel cells, making them ideal for vehicle and transport applications in addition to stationary applications. These are new developments and research efforts are focused on increasing viability by decreasing cost and increasing efficiency [1,2].

Owing to their excellent proton conductivity below 100 °C, humidified perfluorosulfonic acid membranes such as Nafion^®^ have received much research in fuel cell applications. These membrane materials have good thermal and chemical stability, but they also have several drawbacks that have limited their commercial application, such as difficult external humidification, expensive material costs, and high methanol crossover [3]. Numerous efforts have been conducted to develop new polymer-based materials that can transport protons under anhydrous circumstances in order to avoid these restrictions. Since sulfosalicylic acid (SSA) is a good proton conductor due to its substantial self-ionization and low acid dissociation constant, sulfonation systems using SSA are extensively explored for that purpose. The SSA units primarily offer proton transport via structural diffusion, and PVA was utilized as a host polymer to retain the SSA in their matrix [4]. The miscibility of the blend directly correlates with its final quantity, making miscibility a key consideration in the study of polymer blends [5]. Miscible polymer blends often display one stage, whereas immiscible blends typically display isolated domains. The Gibbs energy of mixing also decreases in miscible polymeric blends [6]. While excellent films were obtained at low polyethylene glycol (PEG) contents, the stage division was observed for polyvinyl alcohol (PVA) blends with polyethylene glycol (PEG) (PVA/PEG) with PEG content higher than 40%. Because PEG and PVA homopolymers mix well, straightforward films were produced for PVA/PEG blends comprising up to 30% PEG [7]. Notwithstanding, the miscibility decreased as the PEG concentration went above that threshold. In PVA/PEG blends, the paradox was observed with higher PEG stacking (above 30%) [8].

The goal of this research is to develop a polymer electrolyte membrane using a casting technique, making it efficient with lower production costs than other membranes produced currently. As a result, in this study, an attempt was made to prepare two groups of samples: (1) blended polymers (PVA/10 wt.% PEG) with various Al_2_O_3_ concentrations; and (2) polymer electrolyte membranes based on PVA/10 wt.% PEG/20 wt.% SSA with various Al_2_O_3_ concentrations, with the goal of using them in fuel cell applications. Ionic conductivity within the samples was also investigated, and nanostructure changes were measured using positron annihilation lifetime spectroscopy.

## 2. Materials and Methods

The polyvinyl alcohol (PVA) with a molecular weight (Mw) of 72,000 g/mol was purchased from Merck Schuchardt, Germany. Polyethylene glycol (PEG) was purchased from Fluka AG, Buchs AG, Switzerland, with the molecular weight of a repeat unit of 40 g/mol. Sigma Aldrich in Germany provided aluminum oxide (Al_2_O_3_ potassium hydroxide (KOH), and sulfosalicylic acid (HO_3_SC_6_H_3_-2-(OH) CO_2_H·2H_2_O). Formaldehyde (CH_2_O) as a solution was purchased from Piochem, Egypt, with an assay of 34–38% and an *M*_w_ of 30 g/mol.

### 2.1. Preparation of Membranes with Sulfosalicylic Acid (SSA)

A clear blended PVA/PEG solution was produced by dissolving 8 g of PVA and 0.888 g of PEG in 72 mL of deionized water at 80 °C while stirring continuously to obtain a clear blended solution of PVA/PEG. Then, to achieve the necessary level of degree of cross-linking (DOC) as described in [9], 4.54 cm^3^ of formaldehyde along with 1 M of KOH (0.448 g of KOH in 8 cm^3^ of deionized water) were added to the blended solution. The degree of cross-linking (DOC) [9] can be calculated as:(1)DOC %=mFMFmPVAMPVA+mPEGMPEG
where *M*_F_ is the molecular weight of formaldehyde; *M*_PEG_, and *M*_PVA_ are the molecular weights for a repeat unit of PVA and PEG, respectively; *m*_F_, *m*_PEG_, and *m*_PVA_ are the weights of formaldehyde, PVA, and PEG, respectively. The ideal value of 60% mole of DOC was attained in this study [9]. The condensation reaction caused the cross-linking between PVA and PEG to happen. As a result, an acetal network was created, and water molecules were released. The mixture was sulfonated by adding 20 wt.% SSA, and the resulting solution was stirred on a hot plate at 80 °C for 10 h to complete the cross-linking reaction. To create the Al_2_O_3_-filled membranes, the mixture was divided into 9 equal parts, and the calculated amount of Al_2_O_3_ (0.0, 0.1, 0.3, 0.7, 1, 3, 5, 7, and 10 wt.%) was then added to the PVA/PEG/SSA blended solutions along with cross-linker. Afterward, the resulting solutions were sonicated long enough to ensure that the Al_2_O_3_ particles were evenly distributed throughout the mixture. The solutions were then placed in Petri dishes and baked for two days at 40 °C. The membranes were removed carefully from the Petri dishes after drying and the membrane thickness was found to be around 59 to 205 µm. The preparation of the materials is shown in Figure 1.

### 2.2. Membrane Characterization Technique

Using a wide-angle X-ray diffraction (WAXD) technique with CuKα (λ = 1.54184 Å) incoming radiation on a Bruker (D8) diffractometer, the crystal structure of the samples under investigation was examined. At room temperature (25 °C) and relative humidity of around 35%, the measurements were made in the 5–80° range.

All the samples were vacuumed for 72 h at room temperature to dry them before the water uptake measurements, and they were then weighted with an electronic balance to record the dry weight for each membrane strip separately. The membrane strips were then thoroughly hydrated by being submerged in deionized water at 30 °C. The strips were taken off after an hour, and the surface was thoroughly dried using water-absorbent paper. Afterward, the mass of each strip was individually recorded. The final process was performed three times to reach the swelling equilibrium before determining the average water uptake, which was calculated as:(2)$$\mathrm{Water}\;\mathrm{uptake}\;\left(\%\right)=\frac{W_{w}-W_{d}}{W_{d}}\times 100.$$
where *W*_wet_ is the weight of the hydrated membrane and *W*_dry_ is the weight of the membrane in the dry state [10].

By exchanging sodium ion (Na^+^) for hydrogen form (H* at first added to the resin), exchange capacity can be calculated. A typical sodium hydroxide solution is then used to titrate the hydrogen ion. The exchange reaction can be represented as:(3)$$2R-H^{+}+2{\mathrm{Na}}^{+}\;(\mathrm{aq})+\mathrm{SO}4^{-2}\;(\mathrm{aq})\;⇉\;2R-{\mathrm{Na}}^{+}+2{\mathrm{Na}}^{+}\left(\mathrm{aq})+\mathrm{SO}4^{-2}\;(\mathrm{aq}\right)$$

The ion exchange resin is denoted by R in the equation. The reaction can be induced to finish by employing a “concentrated” sodium sulfate solution because it is an equilibrium process (as indicated by the double-headed arrow, ⇉). The titration method has been used to determine the IEC. The PVA/PEG/SSA/Al_2_O_3_ membrane samples were dried under vacuum for 24 h before each sample was immediately submerged for 24 h in 20 mL of a 3 M NaCl solution; afterward, 10 mL of the solution was then titrated with 0.05 M NaOH. Drops were added to the solution until the phenolphthalein endpoint, which is when the color of the solution changed from colorless to pale pink. The following equation [11] was used to determine the IEC from the titration findings in units of mmol of NaOH per gram of polymer or meq/g as:
IEC = (0.05 × n × *V*_NaOH_)/*W*_dry_,(4)
where n is the factor corresponding to the ratio of the amount of NaCl used for the immersion of the polymer to the amount used for the titration, which is 2, and *W*_dry_ is the weight of the dry membrane. *V*_NaOH_ (mL) is the volume of the 0.05 M NaOH solution used in the titration until it reaches the endpoint.

By using AC impedance spectroscopy measurements, the proton conductivity of the hydrated membrane strips at various Al_2_O_3_ concentrations was determined. Using a HIOKI-3532-50 LCR Hi-Tester, the AC impedance was measured in the frequency range from 50 Hz to 100 kHz with an oscillating voltage between 50 and 500 mV. The impedance spectra were used to determine the membrane resistance *R*_s_. Point-to-point and through-plane ionic conductivity measurements are the two forms of conductivity measurements. The following relation was used to compute the through-plane ionic conductivity:(5)$$\sigma =\frac{L}{R_{s}A}$$
where *A* is the electrode blocked area (1.77 × 10^−4^ m^2^) and *L* is the thickness (m) of the polymer membrane. After utilizing wipes to carefully wipe away the water from the surface of the membranes, the membrane thickness was tested under hydrated conditions. A millimeter was used to measure the thickness at several locations on the membrane, and the results showed that the average thickness ranged from 59 to 205 µm.

The positron source was created by applying a few drops of a ^22^NaCl carrier-free solution (activity approximately 20 μCi) to a thin Kapton^®^ foil (7 μm thick and 10 × 10 mm^2^), drying it, and then covering it with another foil of similar size. The Kapton^®^ foil had a 10% positron adsorption rate, which was adjusted in the study of the analysis of lifetime spectra and is mostly contributed the short lifetime components. Two vaguely organized films of about 10 × 10 mm^2^ sandwiched the source. By that point, the sandwich had been inserted between the two PAL detectors in the sample container. The PAL estimation was performed using a traditional fast–fast coincidence method with a time resolution of 250 ps (full width at half maximum, FWHM). The PAL spectra with more than 1.2 × 10^6^ counts for each sample were obtained at 30 °C, and they were then analyzed using the PALSfit3 [12] program in three lifetime components with variances of fit ≤1.2.

## 3. Results and Discussion

### 3.1. Wide-Angle X-ray Diffraction (WAXD)

The WAXD spectra for PVA/10 wt.% PEG/20 wt.% SSA and different concentrations of Al_2_O_3_ are shown in Figure 2. It is well known that the pure PVA film is semicrystalline in nature, having a characteristic WAXD peak centered broadly near 2θ = 19.5° [13]. According to the Bragg equation [14], the peak position in the WAXD graphs provides information about the change in interlayer spacing, or the d-spacing of the latex nanocomposites:
nλ = 2d sin (θ),(6)
where d is the distance between layers, n is an integer, θ is the angle of diffraction, and λ (=1.54184 Å) is the wavelength. Therefore, the interlayer spacing is approximately 4.55 Å at the mean peak at about 19.5°. Polymers typically exhibit semicrystalline characteristics, lying between amorphous and crystalline states. Small crystalline zones (crystallites) are typically broader by amorphous polymer regions. The deconvolution of the WAXD pattern in the 2θ range from 15° to 25° shown in Figure 2 can be used to determine the relative crystallinity and amorphous regions. The figure makes it evident that all the current membranes under study are semicrystalline and that their degree of crystallinity is between 20 and 39%. The red peak in the figure represents crystalline matter, while the blue peak represents amorphous matter.

The grain size *D* was calculated by applying the Scherrer equation [15]:(7)$$D=\frac{0.9\lambda }{\beta \cos\left(\theta \right)}\;.\;$$
where *λ* is the wavelength, and *β* is the full width at half maximum (FWHM). Table 1 lists the peak position, FWHM, grain size, and area under the peak for crystalline and amorphous peaks. With the addition of Al_2_O_3_ particles, the relative sharpness of the diffraction peaks naturally increases. Figure 3 illustrates a potential interaction between PVA and Al_2_O_3_. As demonstrated in Figure 4, the interaction with alumina causes a clear decrease in the PVA crystallinity [16] with an increase in the Al_2_O_3_ [17] concentrations up to 0.7 wt.%. The degree of crystallinity then rises with an increase in the Al_2_O_3_ particle concentration until leveling off at about 25% at 10 wt.% Al_2_O_3_. A polymer’s chain is more regularly aligned as it becomes more crystalline, which raises its density and hardness. This outcome is consistent with the data from Li et al. [18], which demonstrated that an increase in the Al_2_O_3_ nanoparticle content decreased the degree of crystallization in the polymer and increased the amorphous phase of the membrane by increasing the membrane’s amorphous phase. Additionally, Piotrowska et al. [19] treated the membranes with Al_2_O_3_ nanoparticles, which resulted in a reduction in the number of accessible pore spaces, which in turn decreased the ability to absorb liquid phases. The availability of the alumina filler in an imidazolium-based gel electrolyte was evaluated by Egshira et al. [20].

By offering additional routes for Li^+^ ion movement and altering how the Li^+^ ions interact with the polymer matrix, the Al_2_O_3_ filler improved Li^+^ ion mobility. Additionally, Rai et al. [21] created PVA composite gel electrolytes that were nano-Al_2_O_3_-filled. They discovered that while the PVA membrane has a porous structure, adding 2 wt.% Al_2_O_3_ nanoparticles reduced the porosity of the PVA composite electrolyte because the Al_2_O_3_ nanoparticles are trapped in the pores amid the chains. The PVA chains are completely coated in Al_2_O_3_ nanoparticles after the addition of 6 wt.% Al_2_O_3_ nanoparticles, indicating complete Al_2_O_3_ nanofiller dispersion in the electrolyte film. The grain sizes and shapes become erratic with more Al_2_O_3_ nanoparticle addition (10 wt.%), producing a partly crystalline structure with Al_2_O_3_ nanoparticles and PVA electrolyte. The amorphous phase of pure PVA increased as the Al_2_O_3_ nanoparticle content increased. Therefore, the increased crystallization was caused by the polymer chain diffusion and nucleation rates.

The intensity decreases in the PVA peak in the membranes following Al_2_O_3_ particle embedding is another intriguing finding from WAXD data. This is explained by the breakdown of the hydrogen bonding between the hydroxyl groups on PVA chains and the embedded Al_2_O_3_ particles (Figure 3). As a result, molecular chains are significantly freer to spin [22]. The differentiating peaks for Al_2_O_3_ particles at concentrations less than 5% by weight would not be reflected in this structural analysis result using the PVA/PEG/SSA/Al_2_O_3_ particle WXRD spectra (Figure 2). Additionally, the strong interaction between PVA and Al_2_O_3_ is to blame for the disappearance of the distinctive Al_2_O_3_ peaks (ASTM 75-1862) in low Al_2_O_3_ concentrations and the reduced crystallinity of the PVA-Al_2_O_3_ nanocomposite (Figure 3) [13,23]. However, the peaks at 2θ = 39.3° and 67.2° that are connected to the X-ray spectra for Al_2_O_3_ can be seen in the X-ray spectra for large concentrations of Al_2_O_3_ [24]. One of the primary causes is the relatively small number of Al_2_O_3_ particles compared to the amount of bulk PVA. As a result, the predominance of the amorphous character contributes to scattering, which was seen in the spectra of PVA/PEG/SSA/Al_2_O_3_ and in the data of Abdullah and Saleem [25]. As a result, there is no chance of precisely determining Al_2_O_3_’s contribution to the entire scattering spectrum.

### 3.2. Water Uptake

The hydrophilicity and fouling qualities of the produced polymeric membranes were improved when components such as aluminum oxide (Al_2_O_3_) nanoparticles were used in their synthesis [26,27]. Because of a condensation reaction, PVA and PEG became cross-linked. According to the suggested reaction plot (Figure 3), this resulted in the arrangement of an acetal linkage network and the release of water molecules. The proposed reaction plot demonstrated the intermolecular cross-linking among PVA and PEG molecules. Note that Figure 3 shows that alumina is similarly associated covalently and, thus, Al-O and Al-OH bonds are framed. Figure 5 shows the water uptake as a function of Al_2_O_3_ concentrations on the polymer membranes without and with SSA. The water uptakes for the polymer composite without SSA are lower than those for the membrane with SSA, indicating that the SSA leads the membrane to absorb more water. This is consistent with research from Elsharkawy et al. [28], who discovered that increasing the ion exchange capacity IEC for per-fluorinated sulfonic acid/PTFE proton exchange membranes increases the water uptake. In addition, it is clear from Figure 5A for the group without SSA that the water uptake decreases with an increase in the Al_2_O_3_ concentrations up to 5 wt.% and then increases with an increase in the Al_2_O_3_ concentrations, indicating increasing the hydrophilicity of the polymer with more Al_2_O_3_ filler. For the second group that includes SSA, as shown in Figure 5B, the behavior is opposite compared to the other group (without SSA). It is clear from the figure that the water uptake decreases with an increase in the Al_2_O_3_ concentrations up to 0.7 wt.%, increases with an increase in the Al_2_O_3_ concentrations up to 3 wt.%, and then decreases with an increase in the Al_2_O_3_ concentrations, demonstrating a decrease in hydrophilicity with more Al_2_O_3_ particles. This is because inclusion of alumina restricts the mobility of the polymeric chains, which ultimately causes a reduction in the membranes’ ability to absorb water [29]. This demonstrated that the polymers in the membranes were cross-linked. It is noteworthy that the behavior of the water absorption in the presence of Al_2_O_3_ may be related to altering the level of crystallinity, as previously discussed.

### 3.3. Ion Exchange Capacity

Ions bound to a high molecular weight polymer are exchanged for other ions in solution during the ion exchange process. Normal forms of the high molecular weight polymer are tiny, sphere-like objects known as beads. Ion exchange resins are employed in the demineralization of water and ion mixture separation processes. Ion exchange capacity (IEC), measured in milliequivalents of exchangeable ions per gram of resin (meq/g), is one of the most crucial properties of ion exchange resin. In the present work, the IEC of PVA/PEG/SSA/Al_2_O_3_ membranes is almost constant at about 0.8 meq/g, which indicates there is no effect from changing the Al_2_O_3_ concentrations in PVA/PEG/SSA polymers. The value obtained is near the IEC of Nafion NR212 (0.9 meq/g) [30].

### 3.4. Ionic Conductivity

Figure 6 shows the Cole–Cole plots for the PVA/10wt.% PEG and different concentrations of Al_2_O_3_. Two distinct regions can be seen in the plots, a low-frequency region inclination spike and a high-frequency region semicircle arc. The semicircle in this study is a representation of the samples’ bulk effects and results from the parallel coupling of their bulk capacitance [31] and bulk or sample resistance [*R*_s_]. The intercept between low-frequency and high-frequency regions on the *Z*-axis can be used to determine the value of *R*_s_. Using Equation (5) and the bulk resistance deduced from Figure 6, the ionic conductivity as a function of Al_2_O_3_ concentration on PVA/10 wt.% PEG is presented in Figure 7. It is clear from the figure that the ionic conductivity increases up to 1 wt.% and then decreases with an increase in the Al_2_O_3_ concentrations up to 3 wt.%. Finally, it increases to 58 mS/m at 10 wt.% Al_2_O_3_ concentrations. The semicircle appears only for 3 wt.% Al_2_O_3_ concentration and has the lowest conductivity, as shown in Figure 7. This explains the disappearance of the semicircles in other compositions where they have higher ionic conductivities. This led us to conclude that the charge carriers are ions and that the conductivity for those compositions is mainly due to ions [32]. Ionic transport in the polymers appears to occur on the surface of the Al_2_O_3_ particles or through a low-density polymer phase at the interface, disconnected from the polymer relaxation mechanisms; this behavior of ionic conductivity is revealed in Figure 7. The effect of Al_2_O_3_ particles was not dominant at low loading. On the other hand, with high loading, agglomeration obscured some of the Al_2_O_3_ particle fillers’ active sites. In this scenario, both the segmental motion of the polymer and the active hops along the filler’s surface allow ions on the polymers to migrate [33].

The Al_2_O_3_ particles altered the orientation of the polymer chain and decreased its crystallinity [34]. The ionic conductivity in the free volume region may be improved by increasing the amorphous phase of the polymer matrix in the nanocomposite [35]. However, the crystallinity of the nanocomposite increased when the Al_2_O_3_ content was increased by more than 0.7 wt.%, showing that the Al_2_O_3_-induced crystalline polymer nucleation was similar to the data of Kumar et al. [36]. A small Al_2_O_3_ filler particle can increase the conductivity of PVA/10 wt.% PEG samples, which can possibly have a stronger effect on the immobilization of the long polymer chains. However, according to Zhao et al. [37], the fine (nanosized) filler grains would obstruct one another, increasing immobilization and decreasing ionic conductivity.

Figure 8 shows the Cole–Cole plots for the PVA/10 wt.% PEG/20 wt.% SSA membranes and different concentrations of Al_2_O_3_. The plots display two clearly defined zones, a high-frequency region semicircle arc and a low-frequency region inclination spike, as evidenced by the data in Figure 6. Additionally shown in Figure 9 is the proton conductivity as a function of Al_2_O_3_ concentrations in PVA/10 wt.% PEG/20 wt.% SSA membranes using Equation (5) and the bulk resistance calculated from Figure 8. These observations suggest that raising Al_2_O_3_ concentrations to 0.7 wt.% improves the membrane’s proton conductivity. Additionally, a closer look at all the curves shows that the conductivity rises as the amount of Al_2_O_3_ increases, reaching a maximum value of 0.135 S/m when the Al_2_O_3_ content reaches 0.7 wt.%, as opposed to the conductivity of the membrane without Al_2_O_3_ (0.03 S/m). However, the conductivity value decreases in the contrary with further increasing Al_2_O_3_. This proton conductivity behavior is related to the preceding water uptake behavior (Figure 5B), where more water uptake leads to higher proton conduction, especially in sulfonated membranes similar to Nafion [30]. The abovementioned variation in membrane morphology and electrolyte uptake in the first group samples (without SSA) most likely explains the causes. The fact that Al_2_O_3_ provided more H^+^ transport sites on the surface of the membrane is one factor that could explain this observation. The sample PVA-maximal Al’s membrane absorption is presumably the second cause. However, many Al_2_O_3_ aggregates obstruct the transit of H^+^, whereas further Al_2_O_3_ adsorption reduces proton conductivity [38].

### 3.5. Positron Annihilation Lifetime

Figure 10 shows the *o*-Ps lifetime *τ*_3_ and its intensity I_3_ as a function of Al_2_O_3_ concentrations on PVA/10 wt.% PEG polymers and PVA/10 wt.% PEG/20 wt.% SSA membranes. It is clear from Figure 10A that the *o*-Ps lifetime for the PVA/10 wt.% PEG polymers increases slowly with an increase in the Al_2_O_3_ concentrations. This behavior could be connected to the free volume of the polymers. The free volume cavity radius *R* is related to the *o*-Ps lifetime *τ*_3_ by a simple relation according to the Tao–Eldrup model [39,40]:(8)$$\tau_{3}={0.5\left\{1-\frac{R}{R_{o}}+\frac{1}{2\pi }\sin\left(\frac{2\pi R}{R_{o}}\right)\right\}}^{-1}(ns),$$

The underlying assumption in the formulation of this relationship is that the *o*-Ps atom in a free volume cell can be approximated to a particle in a potential well of radius *R*_o_. It is assumed that *R*_o_ = *R* + Δ*R*, where Δ*R* = 1.656 Å represents the thickness of the homogenous electron layer in which the positron in *o*-Ps annihilates [41].

Using the value of *R*, the hole volume size *V*_f_ in nm^3^ is calculable [42,43,44]:*V*_f_ = 4 π*R*^3^/3.(9)

The right axis of the right top of Figure 10 presents the hole volume size calculated using Equations (8) and (9). The relative fractional of hole volume *f*_r_ can be calculated as *o*-Ps intensity multiplied by the hole volume size *V*_f_:
*f*_r_ = *I*_3_ *V*_f_.(10)

The behavior of the hole volume size with Al_2_O_3_ concentrations in PVA/10 wt.% PEG polymers can be associated with the adjustment of the degree of crystallinity. Expanding the hole volume size is because of expanding the amorphous region (i.e., decreasing the crystalline region). The current data are similar to those of Riyadh et al. [17], who discovered that connecting Al_2_O_3_ particles somewhat increased the sharpness of the diffraction peaks, showing that interaction with alumina causes a reasonable decrease in PVA crystallinity. Additionally, the collaboration between PVA and Al_2_O_3_ is credited with the disappearance of the distinctive Al_2_O_3_ peaks and the decreased crystallinity of the PVA-Al_2_O_3_ nanocomposite [23]. The *o*-Ps intensity I_3_ for PVA/10 wt.% PEG polymers did not change with an increase in the Al_2_O_3_ concentrations, which demonstrates the consistency of positronium formation, as shown in Figure 10B.

It is interesting to note that the o-Ps lifetime (*τ*_3_ = 1.772 ns) for PVA/10 wt.% PEG (without Al_2_O_3_) is higher than that for pure PVA (*τ*_3_ = 1.25 ns) [45], indicating an increase in the hole volume size by adding PEG in PVA, in the other words, increasing the amorphous region in the polymer blends. A drop of crystallinity is observed for almost all blends containing PEG. The decrease in the crystallinity is attributed to the reduction in PVA chain mobility and the formation of H-bonding between PEG and PVA during the blending process, which hindered the crystallization step. Similar results have been published showing that PVA/PEG mixes with 10 weight percent PEG had less crystallinity [5]. Figure 10D,E show the *o*-Ps lifetime *τ*_3_ and its intensity *I*_3_ as a function of Al_2_O_3_ concentrations on PVA/10 wt.% PEG/20 wt.% SSA membranes. It is clear from the figure that the *o*-Ps lifetime *τ*_3_ increases slightly at 1 wt.%, decreases up to 3 wt.%, and then increases again with an increase in the Al_2_O_3_ concentrations in the membranes. This behavior agrees with the behavior of the degree of crystallinity (Figure 4), whereas the larger free volume connected to the small degree of crystallinity and smaller size of hole volumes leads to the high degree of crystallinity. On the other hand, the *o*-Ps intensity *I*_3_ did not change with an increase in the Al_2_O_3_ concentrations, indicating the constant of positronium formation similar to the data in Figure 10B. It is noted that the membranes with SSA had low *o*-Ps lifetime *τ*_3_ (i.e., small hole volume size) and high *o*-Ps intensity *I*_3_ compared to the polymers (without SSA). The data for *o*-Ps intensity *I*_3_ are opposite to what is expected from SSA. It is well known that the sulfonation of the membrane decreases the *o*-Ps intensity because of inhibition or increasing the amorphous region in the sample. It is clear from Figure 10C,F that the relative fractional hole volume *f*_r_ is smaller for the membrane with SSA than that for the polymer without SSA. This agrees with the data of Gomaa et al. [12], who found that the hole volume contents are decreased with an increase in the amount of SSA. Adding more SO_3_ groups into the polymer chains resulted in the modification of the chain conformation and packing.

Understanding the properties of polymers requires good comprehension of hole volume. The hole volume is generally generated from the irregular dispersion of holes in polymer atomic chain fragments. The hole volume is answerable for the directional conduction of protons in PEM used in fuel cell applications. Controlling the hole volume content of any polymer can change its physical and substance properties. Figure 11 shows the effect of free volume on water uptake and ionic conductivity. It is shown in Figure 11A that the water uptake of the membranes (with SSA) is emphatically relative to the hole volume size that improves the layer’s protection from water assimilation for the membranes. However, there is no rational relationship for polymers that do not contain SSA. Furthermore, the proton conductivity is found to increment with an expansion in the hole volume size, as displayed in Figure 11B, because protons are permitted to take a leap toward an adjoining site when there are sufficiently enormous free volumes encompassing the protons [46]. Additionally, there is no unmistakable relationship between the ionic conductivity and hole volume size for the samples without SSA. Figure 11 represents a strong correlation between the microscopic properties obtained from positron annihilation lifetime spectroscopy results and other macroscopic properties assessed from other techniques such as water uptake and ionic conductivity.

## 4. Conclusions

Cross-linked mixed matrix membranes filled with different concentrations of Al_2_O_3_ were successfully prepared using the solvent casting method employed in fuel cell applications. To analyze the electrochemical and physical properties of the samples under investigation, a variety of parameters were measured. It was concluded that the degree of crystallinity of the membranes was affected by the concentration of Al_2_O_3_, where the low crystallinity was at 0.7 wt.% Al_2_O_3_. It was discovered that the water uptake was SSA-dependent, meaning that SSA had an impact on the hole volume size in the membranes in addition to the Al_2_O_3_ group concentration (SSA content). The ionic conductivity of the materials was increased by SSA to a high value (0.13 S/m) at 0.7 wt.% Al_2_O_3_. In addition, the ionic conductivity of polymers without SSA tends to increase with an increase in Al_2_O_3_ concentration. The results of the positron annihilation lifetimes revealed that as the Al_2_O_3_ concentration increased, the hole volume content was enhanced for the polymer without SSA. However, it was densified for the membrane with SSA. A great effect of the Al_2_O_3_ content on the chemical and physical properties of cross-linked PVA/PEG without and with SSA was observed and investigated through different characterization procedures. According to the findings of the study, PVA/PEG/SSA/0.7 wt.% Al_2_O_3_ could be used successfully as a PEM with high proton conductivity for fuel cell applications.

## Figures and Tables

**Figure 1 polymers-14-04029-f001:**
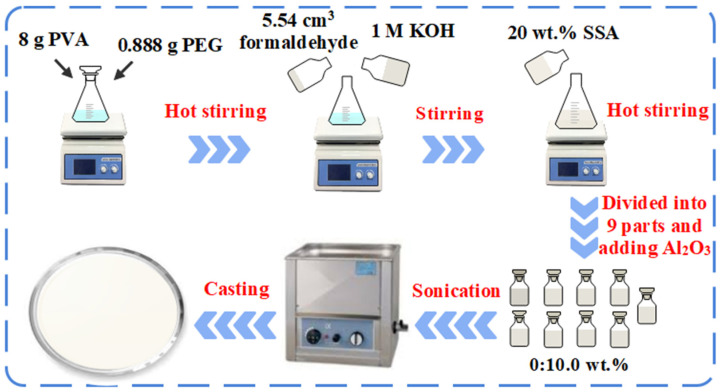
Preparation of PVA/PEG/Al_2_O_3_ and PVA/PEG/SSA/Al_2_O_3_ samples.

**Figure 2 polymers-14-04029-f002:**
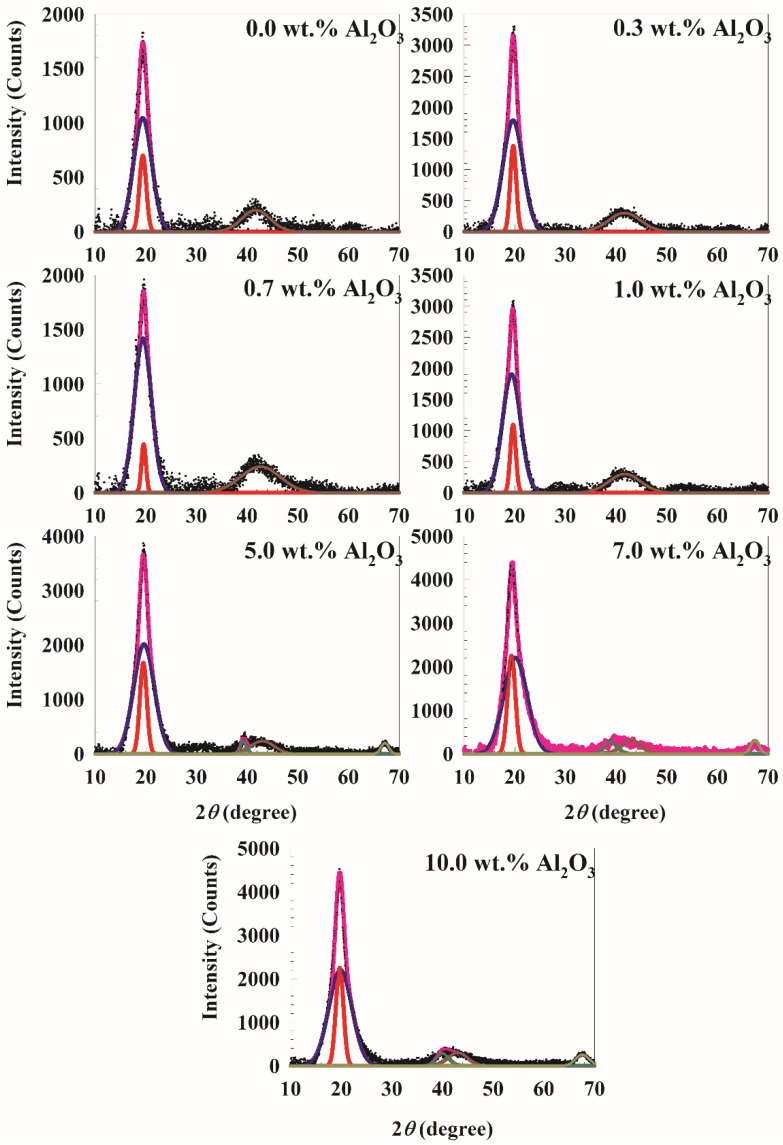
WAXD spectra for PVA/10 wt.% PEG/20 wt.% SSA membranes with different concentrations of Al_2_O_3_. The red peak presents the crystalline matter while the blue peak presents the amorphous matter in the PVA. In addition, dark yellow and dark red represent the peaks for Al_2_O_3_.

**Figure 3 polymers-14-04029-f003:**
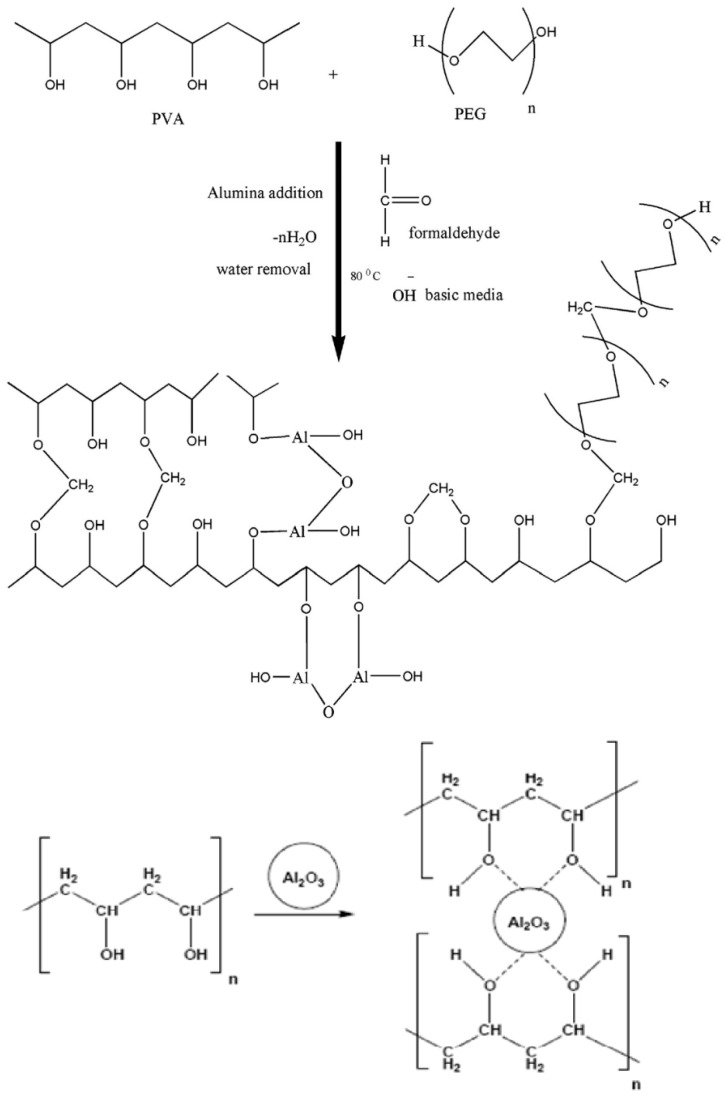
Proposed reaction scheme of alumina-filled cross-linked PVA/PEG polymers.

**Figure 4 polymers-14-04029-f004:**
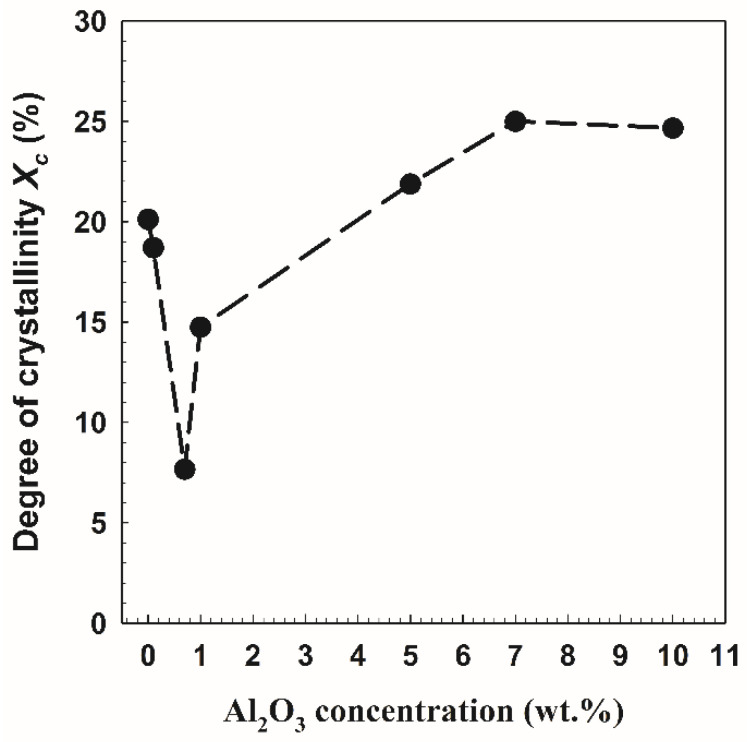
The degree of crystallinity as a function of Al_2_O_3_ contents in PVA/10 wt.% PEG/20 wt.% SSA membranes.

**Figure 5 polymers-14-04029-f005:**
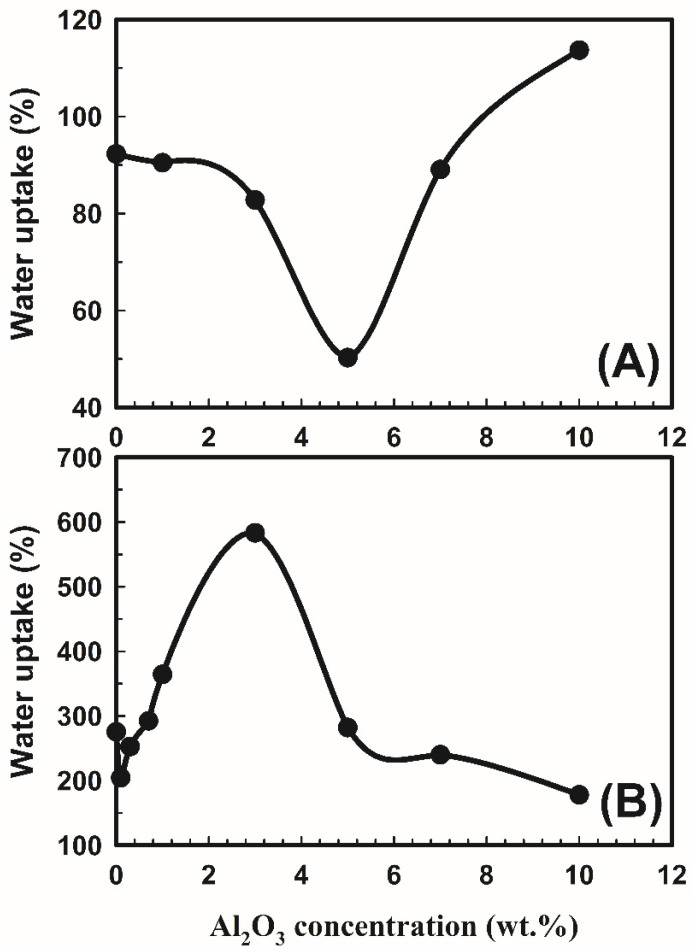
The water uptake for the different concentrations of Al_2_O_3_ on the membrane: (**A**) PVA/10 wt.% PEG polymers and (**B**) PVA/10 wt.% PEG/20 wt.% SSA membranes.

**Figure 6 polymers-14-04029-f006:**
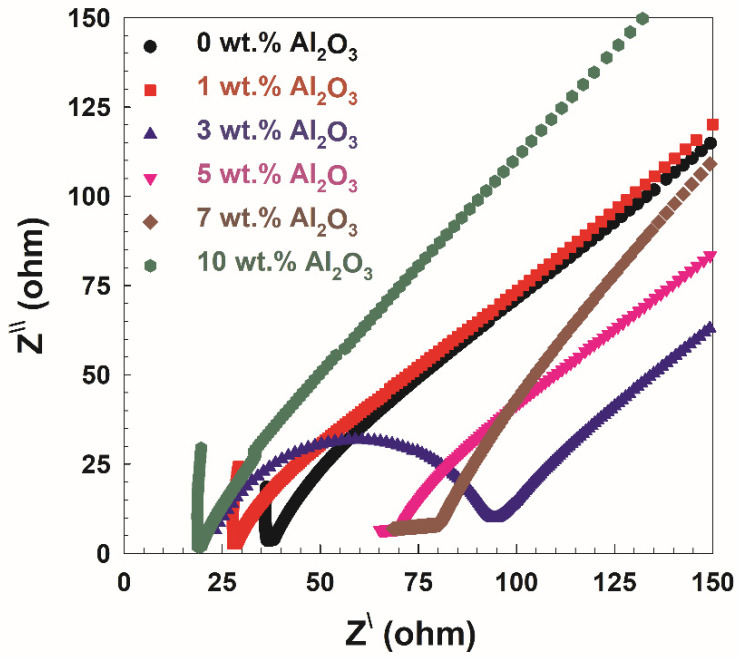
Cole–Cole plot for PVA/10 wt.% PEG polymers and different concentrations of Al_2_O_3_.

**Figure 7 polymers-14-04029-f007:**
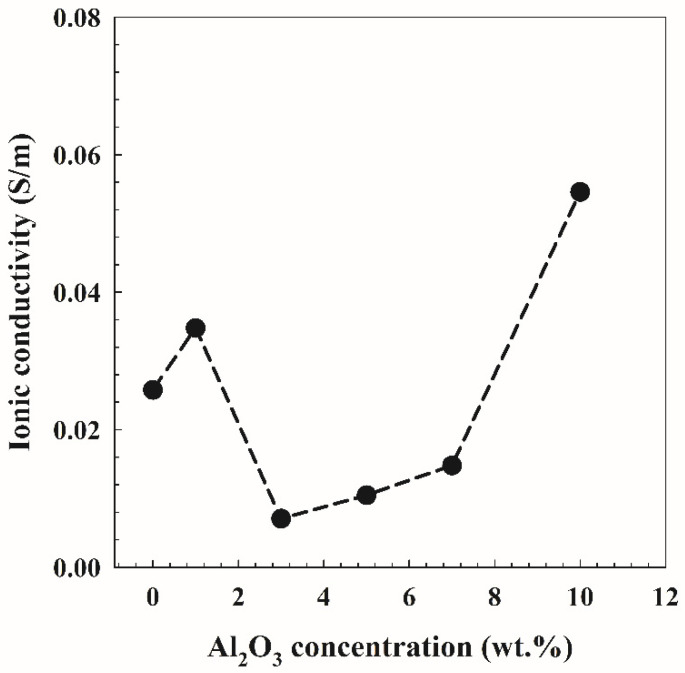
Ionic conductivity as a function of Al_2_O_3_ concentration on PVA/10 wt.% PEG polymers.

**Figure 8 polymers-14-04029-f008:**
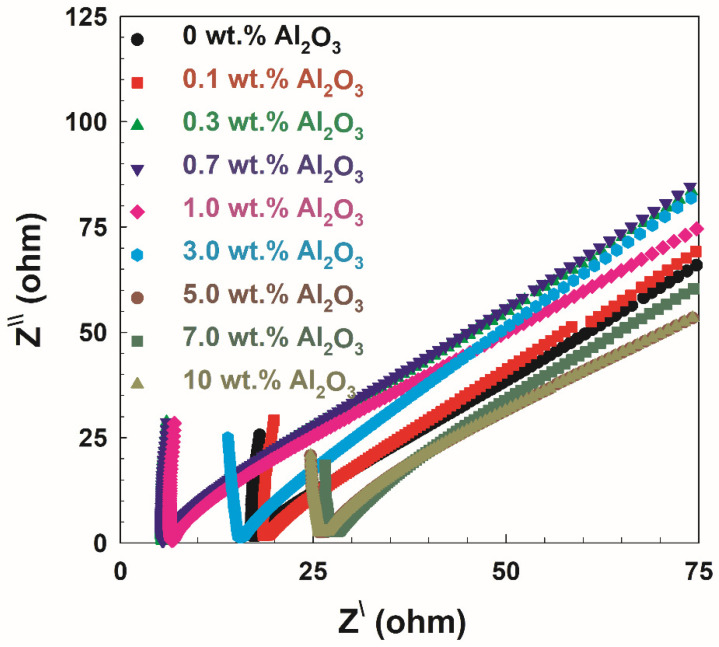
Cole–Cole plot for PVA/10 wt.% PEG/20 wt.% SSA membranes and different concentrations of Al_2_O_3_.

**Figure 9 polymers-14-04029-f009:**
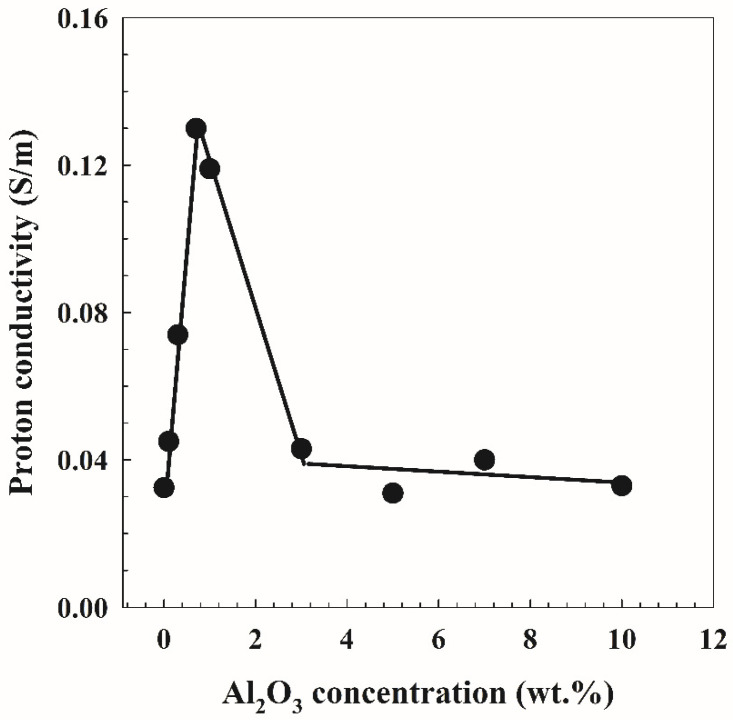
The proton conductivity as a function of the concentrations of Al_2_O_3_ on PVA/10 wt.% PEG/20 wt.% SSA membranes.

**Figure 10 polymers-14-04029-f010:**
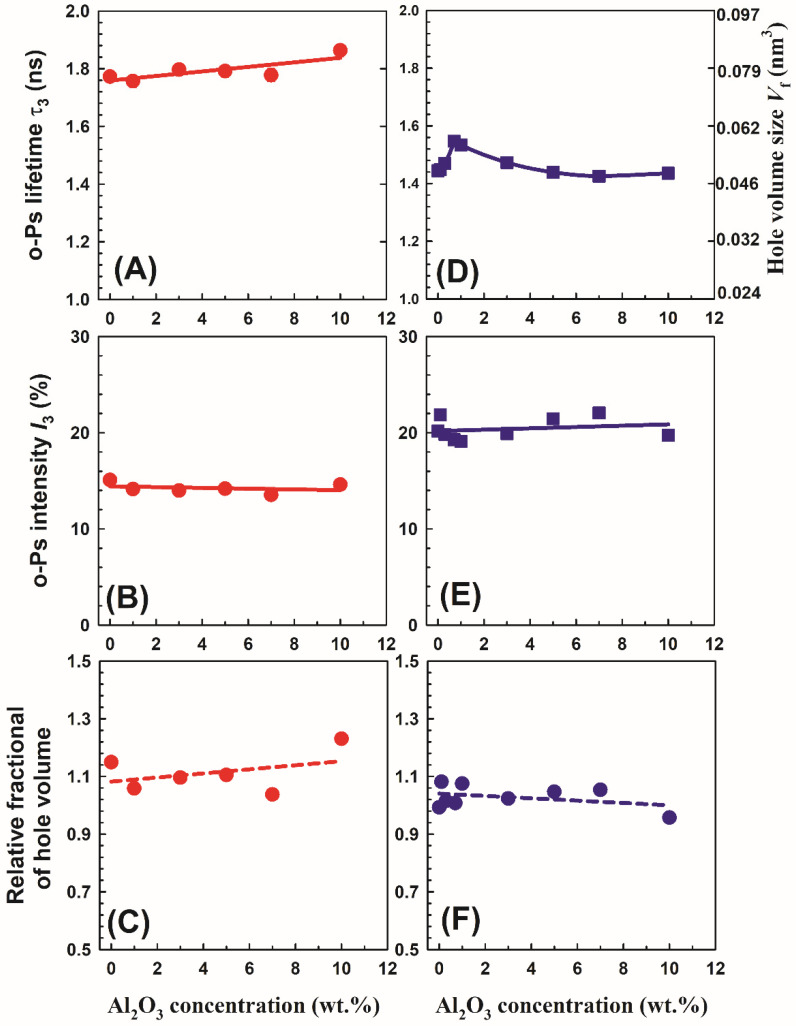
The *o*-Ps lifetime *τ*_3_, its intensity *I*_3_, and relative fractional of hole volume *f*_r_ as a function of Al_2_O_3_ concentrations on PVA/10 wt.% PEG polymers (**A**–**C**) and PVA/10 wt.% PEG/20 wt.% SSA membranes (**D**–**F**), respectively. Included in the right top plot is the hole volume size *V*_f_ calculated using Equations (8) and (9).

**Figure 11 polymers-14-04029-f011:**
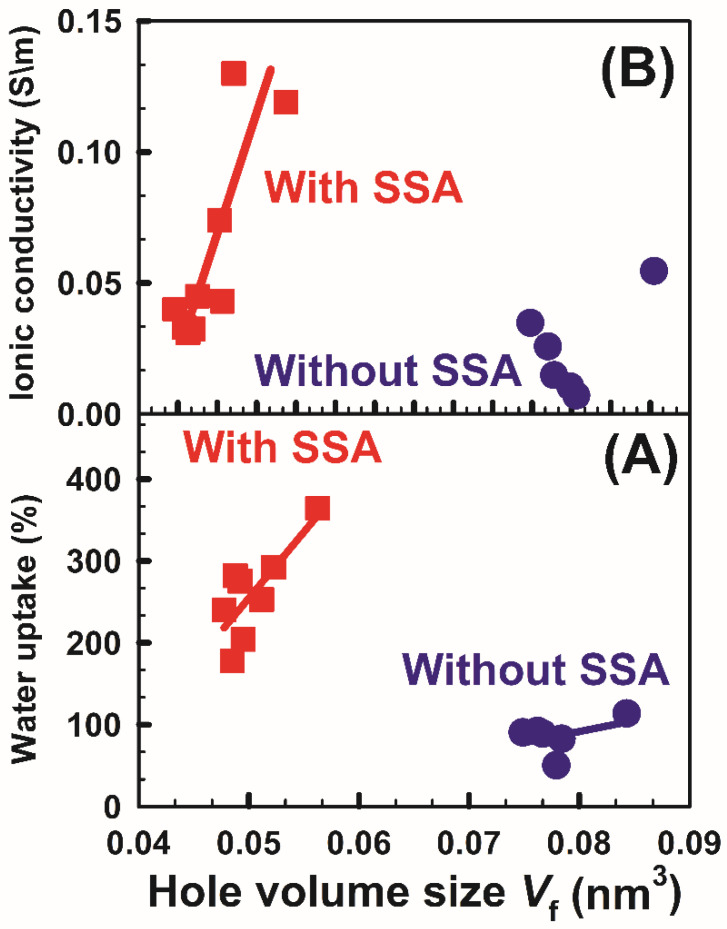
The correlation between the hole volume size *V*_f_ and both (**A**) water uptake and (**B**) ionic conductivity for samples without and with SSA.

**Table 1 polymers-14-04029-t001:** The peak position, FWHM, grain size, and area under the peak for the crystalline and amorphous peaks for PVA/10 wt.% PEG/20 wt.% SSA membranes with different concentrations of Al_2_O_3_.

Al_2_O_3_ Content (wt.%)	Crystalline Peak				Amorphous Peak			
	Peak Position	FWHM	Grain Size (nm)	Area under Peak (%)	Peak Position	FWHM	Grain Size (nm)	Area under Peak (%)
0	19.412°	1.600°	5.26	20.11	19.375°	4.265°	1.97	79.89
0.3	19.704°	1.397°	6.03	18.70	19.655°	4.673°	1.80	81.30
0.7	19.558°	1.087°	7.75	7.67	19.462°	4.078°	2.07	92.33
1	19.673°	1.358°	6.20	14.74	19.366°	4.234°	1.99	85.26
5	19.509°	1.630°	5.17	21.88	19.640°	4.840°	1.74	78.12
7	19.353°	1.812°	4.65	25.00	20.131°	5.465°	1.54	75.00
10	19.700°	1.760°	4.79	24.66	19.837°	5.400°	1.56	75.34

## Data Availability

The authors confirm that the data supporting the findings of this study are available within the article.

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
