# Peer review of "Effect of Al2O3 on Nanostructure and Ion Transport Properties of PVA/PEG/SSA Polymer Electrolyte Membrane"

_polymers, 2022, doi:10.3390/polym14194029_

Round 1

Reviewer 1 Report

The use of Al2O3 nanoparticles as a membrane filler to change their proton conductivity is an interesting idea. As the research results of this work showed PVA/PEG/SSA/0.7 wt% Al2O3 could be used successfully as a PEM with high proton conductivity for fuel cell applications. The presented results are of particular interest to specialists in the field of fuel cell design. The work is well-formed, including a detailed description of the method for obtaining membranes, an analysis of their structure, and data on proton conductivity as a function of the concentration of Al nanoparticles. At the same time, several remarks can be made:

- The quality of some drawings could be improved (for example, figure 4,5), as there is a big difference in the quality of the design of the drawings.

- In Figure 6, only at a concentration of 3% Al2O3, a characteristic Cole-Cole diagram is observed, at other concentrations (at least at this scale) they are absent. This is also quite strange since the gradual evolution of the diagrams is not visible. It is unlikely that the high proton conductivity, which is the reason for the "conduction tails" at low frequencies, abruptly turns off at a given concentration. Most likely, at 3% Al2O3, relaxation with a characteristic Debye behavior appears. Why this so requires further research.

- It would be interesting to see the results of the influence of the size of nanoparticles on the proton conductivity of the obtained membranes.

However, these remarks in no way diminish the scientific significance of the results obtained. I wish the authors to continue their research in this field.

Author Response

Reviewer 1:

The use of Al2O3 nanoparticles as a membrane filler to change their proton conductivity is an interesting idea. As the research results of this work showed PVA/PEG/SSA/0.7 wt% Al2O3 could be used successfully as a PEM with high proton conductivity for fuel cell applications. The presented results are of particular interest to specialists in the field of fuel cell design. The work is well-formed, including a detailed description of the method for obtaining membranes, an analysis of their structure, and data on proton conductivity as a function of the concentration of Al nanoparticles. At the same time, several remarks can be made:

- The quality of some drawings could be improved (for example, figure 4,5), as there is a big difference in the quality of the design of the drawings.

Thank you so much for your comments. The resolution for all the figures was revised to be with 1200 dpi resolution and inserted in the revised manuscript.  

- In Figure 6, only at a concentration of 3% Al2O3, a characteristic Cole-Cole diagram is observed, at other concentrations (at least at this scale) they are absent. This is also quite strange since the gradual evolution of the diagrams is not visible. It is unlikely that the high proton conductivity, which is the reason for the "conduction tails" at low frequencies, abruptly turns off at a given concentration. Most likely, at 3% Al2O3, relaxation with a characteristic Debye behavior appears. Why this so requires further research.

Thank you so much for your comment. The semicircle appears only for 3wt.% Al2O3 concentration and has the lowest conductivity as shown in Fig. (7). This explains the disappearance of the semicircles in other compositions where they have higher ionic conductivities. This led us to conclude that the charge carriers are ions and that the conductivity for those compositions is mainly due to ions [32]. We will do more research on this good point in further work.

- It would be interesting to see the results of the influence of the size of nanoparticles on the proton conductivity of the obtained membranes.

Thank you so much for your comment. This point is so interesting, and we are planning to investigate the effect of the size of the nanoparticles on the proton conductivity in the next research study.

However, these remarks in no way diminish the scientific significance of the results obtained. I wish the authors to continue their research in this field.

Reviewer 2 Report

The authors present a cross-linked poly(vinyl) alcohol (PVA)/poly(ethylene) glycol (PEG) mem-9 branes with varying alumina (Al2O3) content which were synthesized using the solvent solution method. There are some comments toward the research.

1.     The abstract should briefly present the scientific case, not just the study results.

2.     The through-plane ionic conductivity was calculated using equation (5), How about the value of the area in this study?

3.     The WAXD, lines 187 and 188, “the blue peak presents the crystalline while the pink peak presents the amorphous”, but Figure (2) at lines 201 and 211, The red peak presents the crystalline while the blue peak presents the amorphous peaks for the PVA.

4.     Fig. 4 shows the degree of crystallinity as a function of Al2O3 contents in PVA/10wt.%PEG/20wt.%SSA. How about the crystallinity of Al2O3 concentration of 7 and 10 wt.%? 

5.     Fig. 5(B) The water uptakes for the PVA/10wt.%PEG/20wt.%SSA membranes. The highest water uptake occurs at Al2O3 concentration of 5 wt.%, but in lines 269 and 270, claim” … Al2O3 concentration up to 0.7 wt.& and then decreases….”

6.     In Fig. 7 It is suggested to explain why ionic conductivity decrease at Al2O3 concentration of 0.7 wt.% and then increase at Al2O3 concentration of 3 wt.%. 

Author Response

Reviewer 2:

The authors present a cross-linked poly(vinyl) alcohol (PVA)/poly(ethylene) glycol (PEG) mem-9 branes with varying alumina (Al2O3) content which were synthesized using the solvent solution method. There are some comments toward the research.

  1. The abstract should briefly present the scientific case, not just the study results.
  2. The through-plane ionic conductivity was calculated using equation (5), How about the value of the area in this study?

Thank you for your comment. The diameter of the electrode is 1.5 cm, so the area of the electrode is 1.77x10-4 m2.

  1. The WAXD, lines 187 and 188, “the blue peak presents the crystalline while the pink peak presents the amorphous”, but Figure (2) at lines 201 and 211, The red peak presents the crystalline while the blue peak presents the amorphous peaks for the PVA.

   Thank you so much for your comment. The sentence was corrected in the revised manuscript to be “the red peak presents the crystalline while the blue peak presents the amorphous”

  1. Fig. 4 shows the degree of crystallinity as a function of Al2O3contents in PVA/10wt.%PEG/20wt.%SSA. How about the crystallinity of Al2O3 concentration of 7 and 10 wt.%? 

     Thank you for your comment. The WAXD was measured for the concentrations 7 and 10 wt.% and added to the revised manuscript with a discussion of the result. The degree of crystallinity is almost the same for the two samples about 25%.

  1. Fig. 5(B) The water uptakes for the PVA/10wt.%PEG/20wt.%SSA membranes. The highest water uptake occurs at Al2O3concentration of 5 wt.%, but in lines 269 and 270, claim” … Al2O3 concentration up to 0.7 wt.& and then decreases….”

Thank you so much for your comment. The sentence was revised to be “the water uptake decreases with an increase in the Al2O3 concentrations up to 0.7wt.%, increases with an increase in the Al2O3 concentrations up to 3wt.%, and then decreases with an increase in the Al2O3 concentrations, demonstrating a decrease in hydrophilicity with more Al2O3 particles”

  1. In Fig. 7 It is suggested to explain why ionic conductivity decrease at Al2O3concentration of 0.7 wt.% and then increase at Al2O3 concentration of 3 wt.%. 

Thank you so much for your comment. The explanation of this behavior was added in the revised manuscript as ‘This proton conductivity behavior is related to the preceding water uptake behavior [Fig. (5B)], where more water uptake leads to higher proton conduct, especially in sulfonated membranes similar to Nafion [30]. “

Reviewer 3 Report

This paper deals with the structural and ionic conductivity of polymer electrolyte comprises Al2O3 nanoparticles and PVA/PEG/SSA. The present paper well-arranged and the results show a significant investigation. Upon careful reading of this manuscript, I do not see any scientific flaws.

Author Response

Reviewer 3:

This paper deals with the structural and ionic conductivity of polymer electrolyte comprises Al2O3 nanoparticles and PVA/PEG/SSA. The present paper well-arranged and the results show a significant investigation. Upon careful reading of this manuscript, I do not see any scientific flaws.